# Improving Route Development Using Convergent Retrosynthesis Planning

**Paula Torren-Peraire** [1 2]   **Jonas Verhoeven** [1]   **Dorota Herman** [1]   **Hugo Ceulemans** [1]   **Igor V Tetko** [2]
**Jörg K Wegner** [3]

## Abstract

Computer-aided synthesis planning approaches have allowed a greater exploration of potential synthesis routes, however, these methods are generally developed to produce linear routes from a singular product to a set of proposed building blocks and are not designed to leverage potential shared paths between targets. These convergent routes allow the simultaneous synthesis of compounds, reducing the time and cost of synthesis across compound libraries. We introduce a novel planning approach to develop convergent synthesis routes, which can search multiple products and intermediates simultaneously, enhancing the overall efficiency and practical applicability of retrosynthetic planning. We evaluate the multi-step synthesis planning approach using extracted convergent routes from Johnson & Johnson Electronic Laboratory Notebooks (J&J ELN) and publicly available datasets and observe that solvability is generally very high across those routes, being able to identify a convergent route for over 90% of the test routes and showing an individual compound solvability of over 98%.

## 1. Introduction

Compound synthesis is a crucial starting point in early-stage drug discovery to validate hit compounds coming out of a target screening exercise. Exploring the structure-activity relationship (SAR) space involves the identification of a synthesis path typically through a process known as retrosynthesis. Retrosynthesis involves hypothetically breaking down a compound into progressive reactants until a set of purchasable or easily synthesizable compounds is reached (Corey, 1991). Multiple approaches have been developed to address retrosynthesis from a machine-learning perspective, which aims to aid experts in the task of selecting a sequence of chemical reactions that can be applied to a compound to arrive at commercially available starting materials.

Within retrosynthesis approaches, there are two main aspects, the single-step model, which suggests which reaction is most relevant to a molecule, and the multi-step synthesis planning algorithm which guides the search to establish the combination and order of the reactions (Zhong et al., 2023). For the latter, different approaches have emerged to address the expansive search, generally following a heuristic to guide the search with approaches such as proof-number search (Heifets & Jurisica, 2012; Kishimoto et al., 2019), Monte-Carlo Tree Search (Segler et al., 2018; Lin et al., 2020; Ishida et al., 2022) which relies on a combination of exploration and exploitation to explore the search, A* search (Chen et al., 2020; Xie et al., 2022) which takes a global view of the task aiming towards synthesizable molecules, and self-play approaches (Kim et al.; Schreck et al., 2019) which train a learned policy through multiple simulated experiences. However, there is an important caveat with these synthesis planning approaches, medicinal chemists typically work in libraries of compounds, where multiple compounds are designed and synthesized simultaneously to explore the activity space of a target of interest (Dandapani et al., 2012; Brown & Boström, 2018; Seneci, 2018). This library synthesis is not reflected in common multi-step approaches which generally focus on the synthesis of a singular compound rather than the synthesis of multiple compounds via common intermediates leading to convergent routes. Few approaches have explored the mutual synthesis of compounds of interest, commonly as a post-hoc analysis step (Gao et al., 2020; Fromer & Coley, 2023; Pasquini & Stenta, 2023) or by altering the search with select examples (Molga et al., 2019; Xie et al., 2022), differing from running the synthesis search concurrently or producing an extensive showcase of compound libraries.

In this work, we develop a graph-based multi-step approach to identify retrosynthetic routes for multiple compounds simultaneously producing convergent routes. This approach prioritizes routes applicable to all target molecules where

---

[1]In-Silico Discovery, Janssen Research & Development, Janssen Pharmaceutica N.V, Beerse, Belgium [2]Institute of Structural Biology, Molecular Targets and Therapeutics Center, Helmholtz Munich, Neuherberg, Germany [3]In-Silico Discovery, Janssen Research & Development, Janssen Research & Development LLC, Cambridge, US. Correspondence to: Paula Torren-Peraire <ptorrenp@its.jnj.com>.

*Accepted at the 1st Machine Learning for Life and Material Sciences Workshop at ICML 2024.* Copyright 2024 by the author(s).

possible, as well as routes for those compounds which cannot be convergently synthesized. Moreover, to ensure the chemical feasibility of our approach we develop a dataset of convergent routes based on the USPTO dataset (Lowe, 2012) and further compare this to convergent routes identified in Johnson & Johnson Electronic Laboratory Notebook data (J&J ELN). The multi-search approach is guided by a single-step retrosynthesis model, using a fine-tuned Chemformer (Irwin et al., 2022). By additionally implementing batch inference we can produce convergent retrosynthesis routes for up to hundreds of molecules, identifying a singular convergent route for multiple compounds in over 90% of compound sets.

## 2. Methods

### 2.1. Convergent Routes Dataset

Convergent routes are synthesis routes comprised of multiple target molecules resulting from common intermediates, we develop a pipeline to identify and extract these convergent routes from reaction data. Starting from the reaction data we clean and standardize all reactions using RDKit (Landrum) and then identify products and reactants based on the atom-mapping from GraphormerMapper (Nugmanov et al., 2022). The reaction data is then split based on document identifiers so that reactions that were carried out together are considered a joint document. Importantly, the reaction data is not deduplicated at this stage given that the same reaction can occur across multiple documents.

For each document we create a directed graph where the molecules are represented as nodes ($V$) and reactions between molecules are represented as edges ($E$), we defer from adding additional reaction nodes as in previous works (Genheden & Bjerrum, 2022; Mo et al., 2021). The graph is set up from a retrosynthetic standpoint where the children of a node are the reactants required for the synthesis of the parent node. Each reaction from a document is added to the graph by adding the molecules individually as nodes and connecting those with the relevant edges for the reaction. Once all reactions are added to the directed graph, the graph is then traversed to identify weakly connected components, each extracted subgraph is treated as an individual synthesis graph.

The target molecules and building blocks of each synthesis graph can then be identified. Given a node, $v_i$, if $v_i$ has no incoming edges then it will be considered a target molecule since the node is not developed further. If $v_i$ has no outgoing edges then the node will be considered a building block since it does not have any prior reactions. If $v_i$ has multiple incoming edges, from multiple target molecules, then it is considered a common intermediate. Given the focus on convergent routes, all synthesis graphs which do not contain

any common intermediates are discarded. Additionally, in reaction data there are cases in which a single compound was synthesized more than once through different reaction pathways, leading to a cycle within the synthesis graph, in this case, the synthesis graph is discarded since the more optimal reaction path cannot be easily established, ensuring that all synthesis graphs are directed acyclic graphs (DAGs). Lastly, once the cleaned convergent synthesis graphs have been established, we ensure that the target molecules within each graph are not simply stereoisomers and that there are no duplicated graphs across the convergent routes dataset.

### 2.2. Multi-Step Synthesis Planning

The multi-step search is based on a directed graph, containing two types of nodes, molecule nodes and reaction nodes. The search is guided by a single-step model which proposes reactants given a product. The top $K$ proposed reactants are added to the search for each product. The probabilities of the proposed reactants are used to select the top $N$ most promising nodes at each iteration. Promising nodes are selected based on the product of the model probabilities along the linear path between each target molecule and a given end node. End nodes must have no outgoing edges, not form part of the building block set nor be at the maximum route length from the closest target molecule. For each end node, the probabilities are averaged across all target molecules for which a path is present. The top $N$ end nodes with the highest score are considered promising nodes and are followed up by the single-step model. The multi-step search is implemented such that the single-step model can use the native GPU inference setting allowing for faster batch inference of the top $N$ most promising nodes.

When starting a retrosynthetic search, all target molecules are instantiated simultaneously as molecule nodes (Supp Fig. S1). At the first iteration, $K$ sets of reactants are proposed for each target molecule. For each target molecule, $K$ child reaction nodes are created ($deg^+(v_i) = K$). There, from each reaction node, a molecule node is added for every molecule from the proposed reactant set ($R$) ($deg^+(v_j) = |R|$), such that every molecule node will have a maximum of $K$ outgoing edges and every reaction node will have the same number of outgoing edges as the number of molecules in the proposed reactant set. If a molecule node already exists in the search, then the reaction node will be linked to the existing molecule node. In the case that any of the proposed reactants for a reaction node are considered invalid then no molecule nodes will be added to the reaction node. Similarly, if any of the proposed reactants return to a molecule node which already forms part of the path between the molecule node and the target molecule then the molecule nodes will not be added to the reaction node. The same process is carried out until any of the stop criteria are reached, these include maximum time or iterations per molecule or

all potential molecule nodes being explored and flagged as either building blocks or the maximum route length. Both maximum time and iterations are set according to the number of target molecules in the compound library, due to the use of batch inference the total number of iterations is further divided by the batch size.

Once the search is finalized, the proposed routes must be extracted. First, the search graph is pruned to remove any paths which do not end in building blocks since a route which does not end in building blocks cannot be considered solved. The proposed building blocks in the search graph are scored by the product of the probabilities of the single-step model at each reaction step from every target molecule to a given building block, averaged across all target molecules. The building blocks with the highest score are explored first. We parse the search graph to extract the relevant route by identifying the most highly scored linear route from each unexplored molecule to an end node. The process continues until a complete synthesis tree is established. This is carried out until all potential building blocks are explored, or the maximum time limit is reached. The proposed routes are ranked using the product of the single-step model probabilities of each reaction step within the route.

### 2.3. Evaluation

To assess the multi-step planning approach, we create and use a convergent routes dataset. We create two types of convergent routes datasets, based on J&J ELN (Neves et al., 2023) data and USPTO (Lowe, 2012) data. For the USPTO dataset, we use the full reaction dataset, including both applications and grants subsets. In the case of USPTO, we assume that all data shows positive yield, for J&J ELN data we select only reactions with yield $\geq 5\%$ to ensure previously successful retrosynthetic routes. We create a convergent routes dataset for each reaction dataset, using the project identifier for J&J ELN and patent identifier for USPTO to delimit documents. From J&J ELN and USPTO, we select 500 and 1000 convergent routes from the respective datasets to create a convergent route hold-out test set.

To train the single-step model, we clean and standardize all reactions, defining products and reactants as with the convergent routes dataset. We remove all reactions which form part of routes in the respective convergent routes test set, then deduplicate the remaining reactions, removing any reactions with multiple products. With each reaction dataset, we carry out a random 90%/10%/10% train/validation/test split. We fine-tune the Chemformer (Irwin et al., 2022) model, based on each reaction dataset.

For the multi-step search, we set a maximum of 2 minutes and 300 iterations per molecule, using a maximum route length of 8 steps. Additionally, we set a maximum

of 300 target molecules per convergent search. We consider the target molecules from each convergent route a library of molecules, giving one compound library per convergent route. The building block set is composed of all end nodes across each route from the respective convergent test set, comprised of almost 5000 molecules for J&J ELN and 10,239 molecules for USPTO convergent test sets. We explore 10 molecule nodes of interest ($N$) at each iteration and set the beam size ($K$) of the single-step model to 5 since accurate next reaction steps are commonly found within the top 5 suggestions of the single-step model (Torren-Peraire et al., 2024). All multi-step searches are run on a single Tesla T4 GPU with 8 CPU nodes.

The analysis of the proposed routes is conducted using solvability, accuracy and F1 score as metrics. We address two types of solvability, complete and partial. Complete solvability refers to whether the top-N route is a singular convergent route which jointly synthesizes all target molecules within a search. Partial solvability refers to whether all target molecules feature in at least one route up to top-N, irrespective of whether the compounds are synthesized conjointly. Accuracy assesses if there is an exact match between the proposed route at top-N and the experimentally validated route from the convergent routes test set by comparing the reactions (edges) of both routes. We further analyze the intermediate accuracy, which scores whether we identify the same common intermediate in both the proposed and experimentally validated route. Accuracy is a very stringent metric particularly in retrosynthesis given that a slight change in a molecule e.g. a different halogen, will lead the route to be deemed inaccurate. F1 score can be used to quantify the similarity between the proposed routes and the experimentally validated route, we combine two F1 scores, based on the reactions (edges) and the molecules (nodes). In the case of the reaction F1 score true positives are defined as reactions that are correctly identified in the route, false negatives are reactions that are not present in the proposed route compared to the experimentally validated route and false positives are reactions that are present in the proposed route but not the experimentally validated route. The same concept applies to the node F1 score with the exception that the target molecules are not included in the comparison to avoid positively skewing the metric towards short routes with multiple target molecules.

## 3. Results & Discussion

### 3.1. Convergent Routes Dataset

Using J&J ELN and USPTO data separately we create a convergent route dataset of each reaction dataset. Convergent routes are particularly prevalent in medicinal chemistry, we find that 79% of all reactions from J&J ELN form part of a convergent route, with 85% of all documents containing

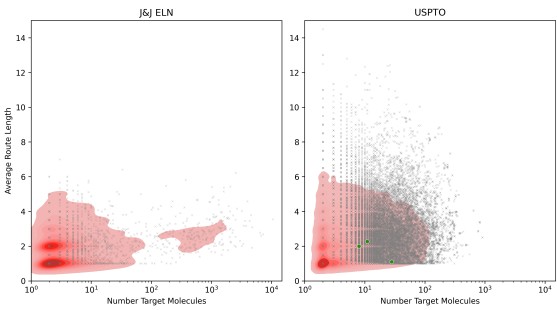

Figure 1. Distribution of the number of target molecules and average route length for convergent routes, per convergent route, from J&J ELN and USPTO. Example routes highlighted in green are shown in Supp Fig. S2.

at least one convergent route. Within USPTO we identify 94,833 convergent routes across all 3.7 million original reactions. We find that 70% of all USPTO reactions are involved in convergent routes with 37% of all documents, patents, containing at least one convergent route. This lower document coverage reflects the skewed distribution of the number of reactions per patent within the USPTO dataset, with 27% of all USPTO patents having only one associated reaction and over half of all projects containing 4 or less reactions (Supp Fig. S3), with previous works (Genheden & Bjerrum, 2022) extracting single molecule retrosynthetic routes, also retrieving a relatively low number of patents.

Convergent routes are generally complex routes, with 61% and 72% of J&J ELN and USPTO routes having more than 2 target molecules and more than 2 reaction steps depth (Fig. 1). Across both J&J ELN and USPTO, most convergent routes have a single common intermediate across all target molecules (Supp Fig. S4). Interestingly J&J ELN convergent routes tend to have a larger number of target molecules whereas USPTO routes tend to be longer in depth. Importantly, convergent routes are often applied to reduce the number of reactions that are necessary to synthesize a set of target molecules, thus also reducing the time and cost of the synthesis. In the case of USPTO, we find that we can reduce the number of reactions required for the synthesis of 988,476 molecules by 40%, going from 2,883,640 reactions when using individual synthesis routes to 1,770,237 reactions through a convergent route approach.

### 3.2. Multi-step search

We develop a new multi-step synthesis planning framework which instantiates multiple target molecules simultaneously, with the aim of convergent route development. Using the convergent route datasets developed for J&J ELN and USPTO we can search convergent routes for real compound libraries to assess the utility of the approach. We randomly select 500 and 1000 convergent routes from J&J

ELN and USPTO respectively as the multi-step test set. We train a single-step retrosynthesis model based on the remaining data, fine-tuning the pre-trained Chemformer on each dataset.

Both single-step models show a similar pattern of accuracy across the top N, achieving high accuracy by the top-10, particularly in the case of the J&J ELN-trained model. The J&J ELN single-step model reaches 85% accuracy at top-10 whereas USPTO reaches 75% accuracy at top-10, with this pattern of 10% difference present across all top-N (Supp Fig. S5). In the case of USPTO, we see that the model performance is lower as compared to the evaluation on USPTO-PaRoutes (Torren-Peraire et al., 2024), which undergoes further data preparation steps. We use these models to guide the respective multi-step synthesis planning for each library of target molecules from the J&J ELN and USPTO test sets.

J&J ELN test set routes show a greater complexity than USPTO convergent routes, given that J&J ELN convergent routes tend to have a higher number of target molecules and common intermediates (Supp Table 1). However, USPTO routes tend to have a higher number of building blocks, potentially due to using less advanced molecules as starting points. When applying the compound libraries to the multi-step search, the approach proposes 53 and 81 routes per compound library on average for J&J ELN and USPTO respectively. Interestingly, for each compound library, we have a large variety of potential common intermediates identified, with 23 unique common intermediate molecule combinations for J&J ELN and 30 unique common intermediate molecule combinations for USPTO within the top 100 routes for each compound library. This shows that the proposed routes have a high diversity, producing multiple options for the potential synthesis of the compound library. Within the proposed routes, the highest ranked route tends to have a similar number of building blocks yet a lower number of common intermediates on average, compared to the experimentally validated routes, in the case of J&J ELN (Table 1). In both J&J ELN and USPTO the proposed routes have a higher number of reactions showing that additional steps are used in the proposed routes leading to longer synthesis paths.

Table 1. Average statistics of the highest-ranked proposed retrosynthetic route for J&J ELN and USPTO compound library test sets.

|  | J&J ELN | USPTO |
| --- | --- | --- |
| Fraction solved molecules | 88.6% | 89.7% |
| Common intermediates | 3.1 | 4.2 |
| Building blocks | 6.3 | 9.3 |
| Molecules | 27.9 | 38.1 |
| Reactions | 19.8 | 26.1 |
| Reactants per reaction | 1.4 | 1.5 |
| Target molecules per intermediate | 3.3 | 3.6 |

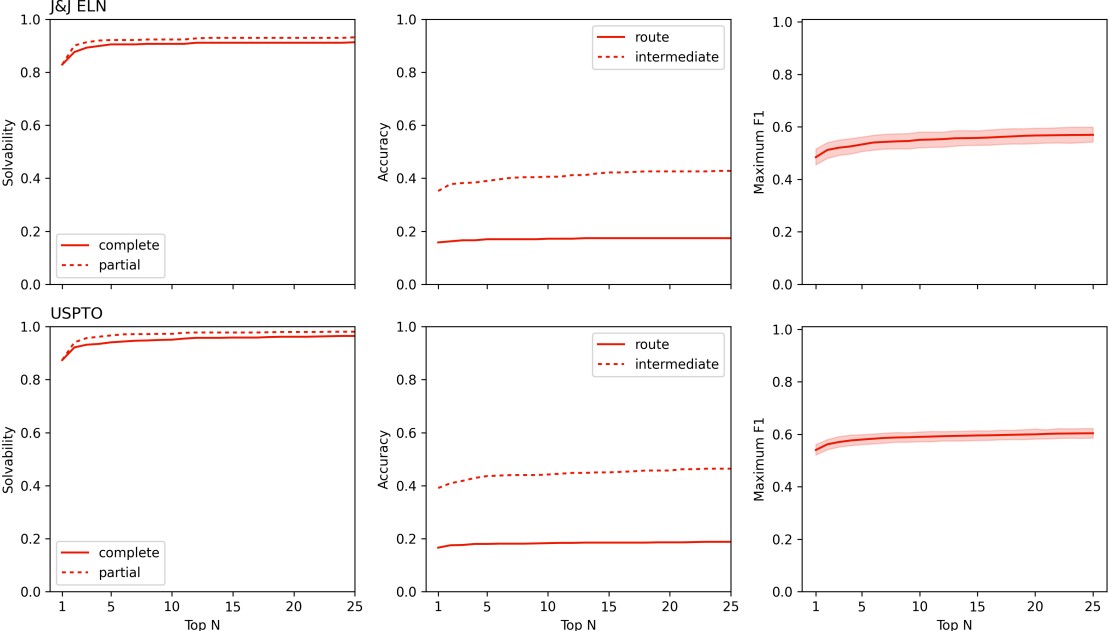

*Figure 2.* Solvability, accuracy and combined F1 score of proposed retrosynthetic routes using J&J ELN (top) and USPTO (bottom) compound library test sets. Accuracy and combined F1 score are calculated compared to the extracted experimentally validated retrosynthetic routes.

Using these libraries of target molecules as test compounds, the multi-step search is almost always able to identify a route to convergently synthesize all target molecules within each library. This shows that the approach can efficiently produce convergent routes within a single search, leading to more streamlined and cost-effective synthesis. Using the convergent route multi-step search approach we can produce a single convergent route for 91.4% of all explored libraries in the case of J&J ELN (Fig. 2), rising to 93.4% when considering solvability for all explored target molecules within each library, irrespective of whether the molecules are synthesized across one or more routes. For USPTO routes, we find a convergent synthesis route for 97.0% of all compound libraries, rising to 98.3% for compound libraries where all target molecules are not identified in the same synthesis route. The small difference between complete and partial solvability shows that the approach can suggest convergent routes in the majority of cases, with a lack of route solvability, i.e. reaching building blocks, being a larger bottleneck than identifying convergence.

Importantly, this approach proposes routes for as many target molecules as possible, irrespective of whether they are conjointly synthesized, such that we can maximize the utility of the retrosynthetic routes. As such, when considering the solvability of individual compounds across all compound libraries, we identify retrosynthetic routes for 98.8% and 99.8% of all individual compounds for J&J ELN

and USPTO respectively. This highlights using convergent routes search to increase the solvability of the multi-step search given that the convergent approach can aid in resolving a greater number of target molecules.

Using the convergent route dataset we can further explore the accuracy of the proposed retrosynthetic routes, a much harder challenge given the numerous alternatives that can be used for compound synthesis (Schneider et al., 2016). We exactly replicate 17.4% of the experimentally validated routes within the top 15 proposed routes for J&J ELN, with a slightly higher accuracy of 18.5% for USPTO routes (Fig. 2). Interestingly, we correctly identify the common intermediate for 42.2% for J&J ELN and 45% for USPTO of the target molecule sets within the top 15 proposed routes. This implies the suggested routes do not follow identical reaction steps as the experimentally validated route however they suggest routes similar to those within the experimentally validated routes. Using the F1 score we can further quantify the similarity of proposed and experimentally validated routes. By calculating and averaging the F1 score of the proposed edges and nodes, we find that over 50% of all libraries have a combined F1 score higher than 0.5 by top-5 (Fig. 2) and more than 36% of routes within the top 15 having a combined F1-score over 0.7 in both datasets.

## 4. Conclusion

Convergent routes, producing the synthesis of multiple target molecules from a shared synthetic path, are a central and common part of medicinal chemistry. Here, we introduce a multi-step synthesis planning approach to develop convergent synthesis routes, which can search multiple products and intermediates simultaneously, enhancing the overall efficiency and practical applicability of retrosynthetic planning, reducing the time and cost of synthesis across compound libraries. We evaluate the multi-step synthesis planning approach using a novel dataset of extracted convergent routes from industry-relevant and publicly available datasets, being able to identify a convergent route for over 90% of the test routes and producing a synthesis route for over 98% of compounds found within the compound libraries. Moreover, the approach shows promising results with the proposed routes being similar to the experimentally validated routes in over a third of the compound libraries.

## Acknowledgements

This study was partially funded by the European Union's Horizon 2020 research and innovation programme under the Marie Skłodowska-Curie Actions grant agreement "Advanced machine learning for Innovative Drug Discovery (AIDD)" No. 956832.

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
