# Supplementary Information

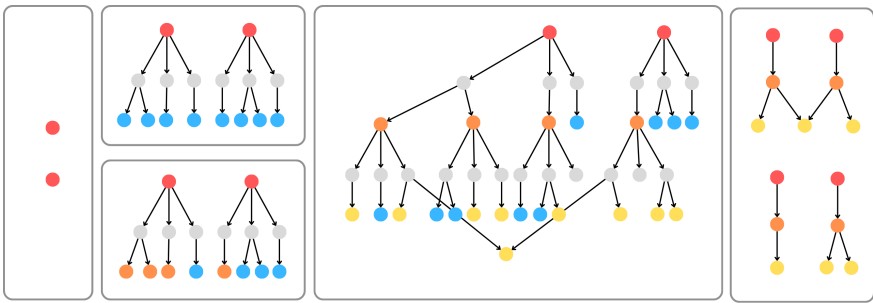

Figure S1: Multistep search process. All target molecules (red) are initiated simultaneously. At the first iteration $K$ sets of reactants are proposed for each target molecule, for each set of proposed reactants a reaction node (grey) is added followed by the relevant reactants (blue). The $N$ most promising molecule nodes (orange) are followed up and K sets of reactants are proposed for each, with the same process continued iteratively until all end nodes are building blocks (yellow) or the time or iteration limit is reached. Molecule nodes which are considered invalid are not added to the search graph. The retrosynthetic routes are then extracted from the search graph. The hypothetical example shown consists of two target molecules, with 3 sets of reactants proposed for each molecule node ($N$) and 4 molecule nodes followed up ($K$) with a maximum of two iterations. Only a sample of the potential extracted routes is shown.

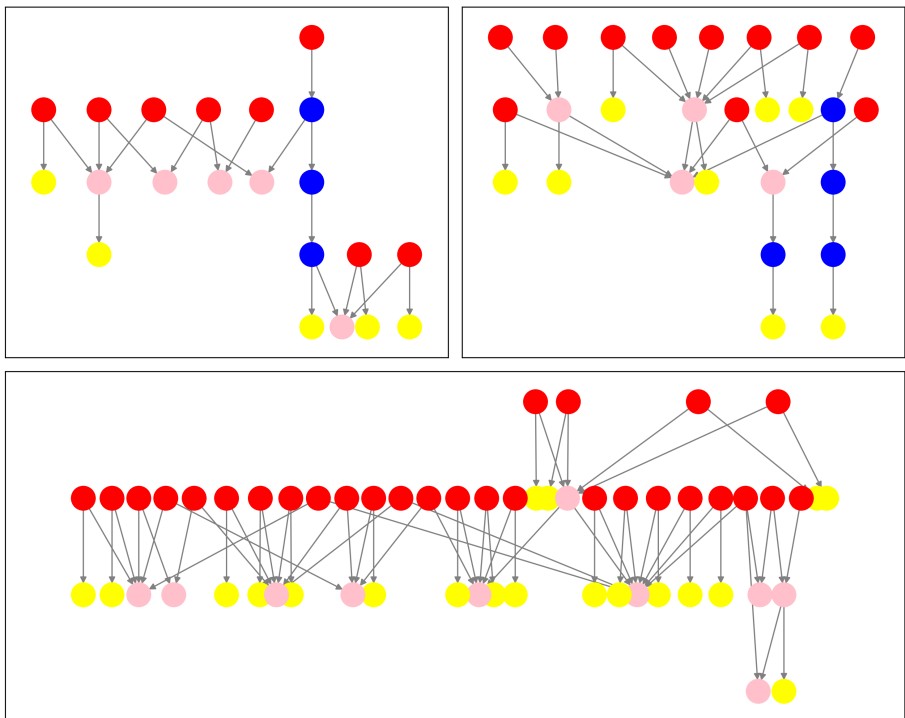

Figure S2: Example convergent routes from USPTO, red nodes show target molecules, pink nodes are intermediates that are used more than once, blue nodes are intermediates that are only used once and yellow nodes show building blocks. The shown routes are highlighted in green in Fig. 1.

Table S1: Average statistics of experimentally validated routes for J&J ELN and USPTO convergent route test datasets

|                                   | J&J ELN | USPTO |
|-----------------------------------|---------|-------|
| Target molecules                  | 12.5    | 10.0  |
| Common intermediates              | 4.9     | 4.0   |
| Building blocks                   | 6.4     | 7.6   |
| Molecules                         | 27.2    | 27.8  |
| Reactions                         | 17.8    | 17.6  |
| Reactants per reaction            | 1.6     | 1.7   |
| Target molecules per intermediate | 4.0     | 3.8   |

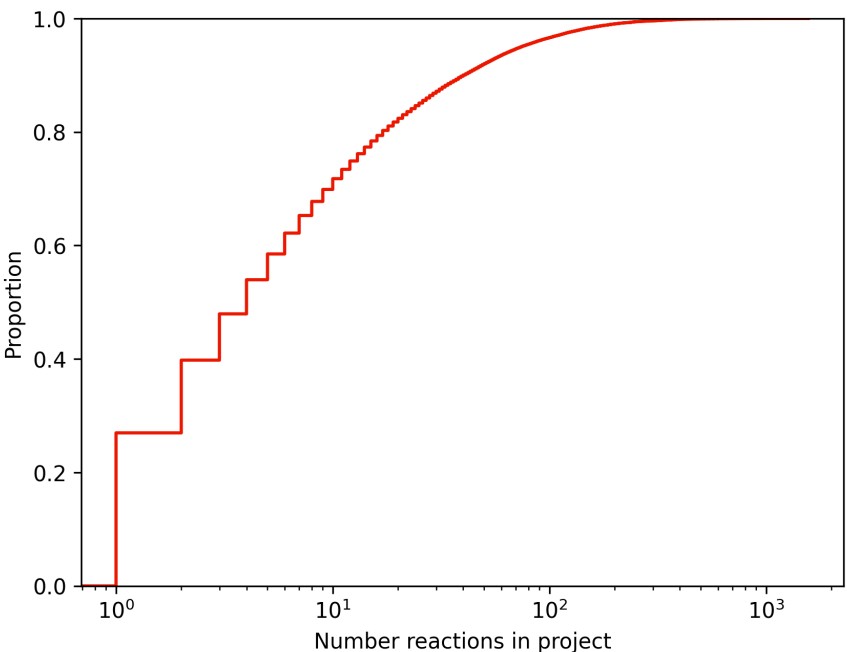

Figure S3: Number of reactions within each USPTO project as a proportion of all USPTO projects.

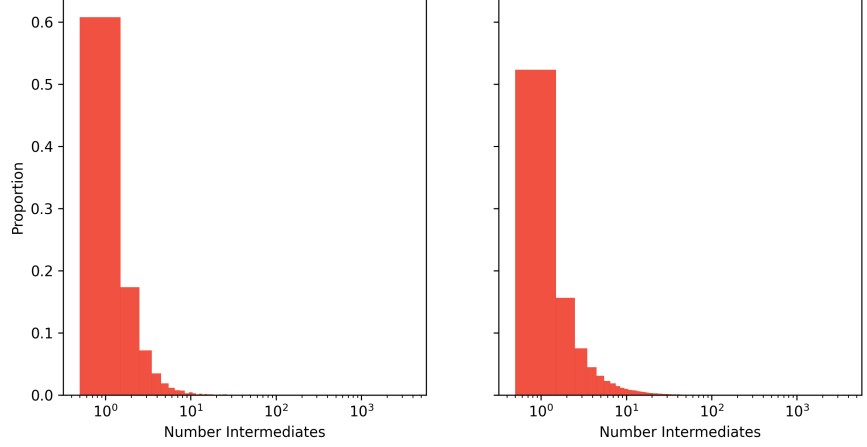

Figure S4: Number of intermediate compounds per convergent route as a proportion of all convergent routes per dataset in J&J ELN (left) and USPTO (right).

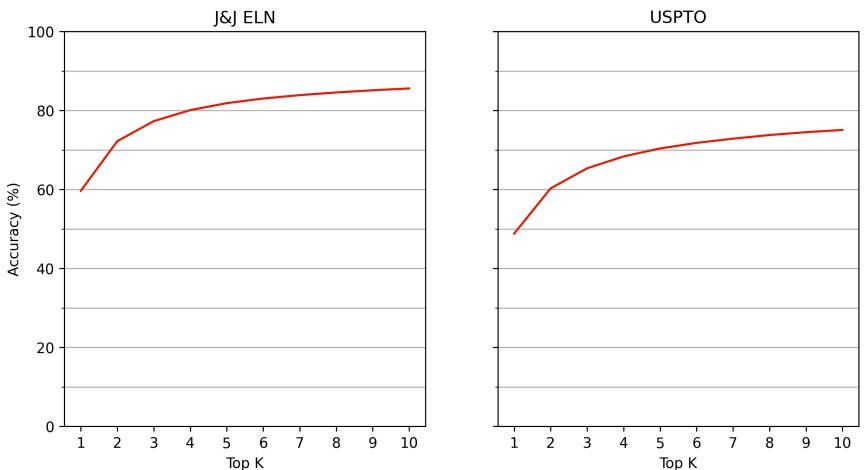

Figure S5: Single-step accuracy of fine-tuned Chemformer for J&J ELN and USPTO within the top K proposed reactants. Accuracy refers to an exact match between experimentally validated reactants and proposed reactants.