# OpenReview forum: "Improving Route Development Using Convergent Retrosynthesis Planning"
_ICML.cc/2024/Workshop/ML4LMS — ML4LMS Poster_

### Official Review · Reviewer_974B · 2024-06-12

**Rating:** 6
**Confidence:** 4

**Review:**

Summary:
The paper presents a new way to design convergent synthesis routes through a multi-step graph based planning approach. The authors fine tune Chemformer to generate reactants from products and score relevant convergent routes in an iterative fashion. The authors follow this up with evaluation and comparison with experimentally validated routes.


Strengths and weaknesses:
The paper presents a novel method towards approaching convergent synthesis route design through a graph-based search aided by a fine-tuned Chemformer model. The datasets used and preparation steps seem to be appropriate for this task. Calculated metrics demonstrate the method's ability to identify convergent routes, improving the efficiency of synthesis tasks compared to individual synthesis routes.

However, as stated by the authors, proposed routes appear to have a smaller number of common intermediates for the J&J ELN dataset along with greater number of reactions / steps for both J&J ELN and USPTO as compared to experimentally validated routes, diminishing practical applicability. Additional work which strengthens the similarity between proposed and experimentally validated reactions would be impactful. A central figure 1 explaining the iterative graph generation along with Chemformer-aided scoring would help readability.

Overall, this paper presents an interesting approach to planning convergent synthesis routes which can drastically improve retrosynthetic workflows. I believe the presented paper aligns nicely with the workshop's theme.

---

### Official Review · Reviewer_HxUd · 2024-06-12
**Review for: Improving Route Development Using Convergent Retrosynthesis Planning**

**Rating:** 6
**Confidence:** 3

**Review:**

-

---

### Official Review · Reviewer_FWQE · 2024-06-12
**Improving Route Development Using Convergent Retrosynthesis Planning**

**Rating:** 7
**Confidence:** 3

**Review:**

The paper introduces a novel retrosynthesis planning approach to model convergent synthesis routes, which can search multiple products and intermediates simultaneously. The authors show that their approach, which employs the Chemformer model for single-step prediction, shows high solvability and is able to identify convergent routes for over 90% of the test routes. Although such an approach is claimed to benefit from improved efficiency and practical applicability, a more rigorous comparison between the proposed method and other retrosynthesis software is lacking. Additionally, an algorithm box and/or a schematic would be helpful for visualisation and to keep track of the pipeline used. Similarly, graphic examples of convergent routes would be helpful. The evaluation on the convergent route dataset could be performed using round-trip accuracy instead of replicating the routes, to account for multiple valid ways of synthesising compounds. The manuscript would benefit from improved clarity and presentation, such as a more schematic explanation the metrics used and the Chemformer model employed, as well as schematics of extracted vs. expected routes etc. Overall, the paper introduces a novel search algorithm with relevance to real chemical synthesis scenarios and contains relevant statistics regarding extracted synthesis routes.